# Phenome-wide Mendelian randomisation analysis identifies causal factors for age-related macular degeneration

**Thomas H Julian[1,2]\*, Johnathan Cooper-Knock[3], Stuart MacGregor[4], Hui Guo[5], Tariq Aslam[2,6], Eleanor Sanderson[7], Graeme CM Black[1,8]\*, Panagiotis I Sergouniotis[1,2,8,9]\***

[1]Division of Evolution, Infection and Genomics, School of Biological Sciences, Faculty of Biology, Medicine and Health, University of Manchester, Manchester, United Kingdom; [2]Manchester Royal Eye Hospital, Manchester University NHS Foundation Trust, Manchester, United Kingdom; [3]Department of Neuroscience, Sheffield Institute for Translational Neuroscience (SITraN), University of Sheffield, Sheffield, United Kingdom; [4]Statistical Genetics, QIMR Berghofer Medical Research Institute, Brisbane, Australia; [5]Centre for Biostatistics, Division of Population Health, Health Services Research and Primary Care, School of Health Sciences, Faculty of Biology, Medicine and Health, University of Manchester, Manchester, United Kingdom; [6]Division of Pharmacy and Optometry, Faculty of Biology, Medicine and Health, School of Health Sciences, University of Manchester, Manchester, United Kingdom; [7]MRC Integrative Epidemiology Unit, University of Bristol, Bristol, United Kingdom; [8]Manchester Centre for Genomic Medicine, Saint Mary's Hospital, Manchester University NHS Foundation Trust, Manchester, United Kingdom; [9]European Molecular Biology Laboratory, European Bioinformatics Institute (EMBL-EBI), Wellcome Genome Campus, Cambridge, United Kingdom

**\*For correspondence:**
thomas.julian@manchester.ac.uk (THJ);
graeme.black@manchester.ac.uk (GCMB);
panagiotis.sergouniotis@manchester.ac.uk (PIS)

## Abstract

**Background:** Age-related macular degeneration (AMD) is a leading cause of blindness in the industrialised world and is projected to affect >280 million people worldwide by 2040. Aiming to identify causal factors and potential therapeutic targets for this common condition, we designed and undertook a phenome-wide Mendelian randomisation (MR) study.

**Methods:** We evaluated the effect of 4591 exposure traits on early AMD using univariable MR. Statistically significant results were explored further using: validation in an advanced AMD cohort; MR Bayesian model averaging (MR-BMA); and multivariable MR.

**Results:** Overall, 44 traits were found to be putatively causal for early AMD in univariable analysis. Serum proteins that were found to have significant relationships with AMD included S100-A5 (odds ratio [OR] = 1.07, p-value = 6.80E−06), cathepsin F (OR = 1.10, p-value = 7.16E−05), and serine palmitoyltransferase 2 (OR = 0.86, p-value = 1.00E−03). Univariable MR analysis also supported roles for complement and immune cell traits. Although numerous lipid traits were found to be significantly related to AMD, MR-BMA suggested a driving causal role for serum sphingomyelin (marginal inclusion probability [MIP] = 0.76; model-averaged causal estimate [MACE] = 0.29).

**Conclusions:** The results of this MR study support several putative causal factors for AMD and highlight avenues for future translational research.

**Funding:** This project was funded by the Wellcome Trust (224643/Z/21/Z; 200990/Z/16/Z); the University of Manchester's Wellcome Institutional Strategic Support Fund (Wellcome ISSF) grant (204796/Z/16/Z); the UK National Institute for Health Research (NIHR) Academic Clinical Fellow and

Clinical Lecturer Programmes; Retina UK and Fight for Sight (GR586); the Australian National Health and Medical Research Council (NHMRC) (1150144).

## Editor's evaluation

The findings of this study as well as the strength of the provided evidence are important and have significance beyond a single subfield. This manuscript is of interest to readers in the fields of ophthalmology, epidemiology and public health. The identification of both known and previously unknown risk factors for age-related macular degeneration (AMD) using genetically informed approaches can be combined with traditional epidemiological approaches to develop interventions that reduce the risk of AMD. The key claims of the manuscript are well supported by the data, and the approaches used are thoughtful and rigorous.

## Introduction

Age-related macular degeneration (AMD) is a common retinal condition that affects individuals who are ≥50 years old. It is caused by the complex interplay of multiple genetic and environmental risk factors, and genome-wide association studies (GWAS) have identified AMD-implicated variants in at least 69 loci. These include important risk alleles in the 1q32 and 10q26 genomic regions (corresponding to the *CFH* [complement factor H] and *ARMS2/HTRA1* locus, respectively) (*Fritsche et al., 2016*; *Winkler et al., 2020*). Other key risk factors include age, smoking, alcohol consumption, and low dietary intake of antioxidants (carotenoids, zinc) (*Chakravarthy et al., 2010*).

AMD can be categorised according to severity (early, intermediate, or advanced) or based on the presence of neovascularisation (neovascular or non-neovascular). Advanced AMD results in loss of central vision, often leading to severe visual impairment (*Fleckenstein et al., 2021*). Notably, AMD is a major cause of blindness in the elderly population and represents a substantial global burden that is expected to continue to grow into the future as an ageing population expands worldwide (*Chakravarthy et al., 2010*).

Mendelian randomisation (MR) is a statistical approach that uses genetic variation to look for causal relationships between *exposures* (such as smoking) and *outcomes* (such as risk of a specific disease) (*Julian et al., 2021*). MR is increasingly being utilised as it can, to a degree, address a major limitation of observational studies: unmeasured confounding (*Sanderson et al., 2022*). To minimise issues with certain types of confounding and to support causal inference statements, MR uses genetic variation as an *instrument* (i.e. as a variable that is associated with the exposure but is independent of confounders and is not associated with the outcome, other than through the exposure). The principles of MR are based on Mendel's laws of segregation and independent assortment, which state that offspring inherit alleles randomly from their parents and randomly with respect to other locations in the genome. A key concept is the use of genetic variants that are related to an exposure of interest to proxy the part of the exposure that is independent of possible confounding influences (e.g. from the environment or from other traits). It is noted that analogies have been drawn between MR and randomised controlled trials with these two approaches considered proximal in terms of hierarchy of evidence (*Julian et al., 2021*). To date, the use of MR approaches in the context of AMD has been limited although these methods have been successfully implemented to explore the relationship between AMD and a small number of traits including lipids, thyroid function, CRP, and complement factors (*Cipriani et al., 2021*; *Han et al., 2021*; *Han et al., 2020b*; *Li et al., 2022*; *Zuber et al., 2020*).

In this study, we developed a systematic, broad ('phenome-wide') MR-based analytical approach and used it to investigate the relationship between early AMD and several thousand exposure variables. A set of traits that are robustly associated with genetic liability to AMD were identified.

## Methods

### Data sources

#### Outcome data

Two AMD phenotypes were used as outcome measures in this study. The first one was early AMD. The GWAS summary statistics for this phenotype were taken from a meta-analysis by *Winkler et al., 2020*. This meta-analysis focussed on populations of European ancestries and used data from the ARIC, AugUR, CHS, GHS, IAMDGC, KORA S4, LIFE-Adult NICOLA, UKBB, and WHI studies (14,034 early AMD cases and 91,214 controls overall). A full description of how these studies classified participants as 'early AMD' can be found in the relevant publication *Winkler et al., 2020*; briefly, a number of approaches considering drusen size/area and the presence or absence of pigmentary abnormalities were utilised including the 3 Continent Consortium (3CC) severity scale (*Klein et al., 2014*), the Rotterdam Eye Study classification (*Korb et al., 2014*), the Beckman clinical classification (*Ferris et al., 2013*), and the AREDS-9 step classification scheme (*Davis et al., 2005*). All relevant studies used colour fundus photography for grading purposes. The second phenotype that we studied was advanced AMD. For this trait, we drew on GWAS summary statistics from a multiple trait analysis of GWAS (MTAG) study by *Han et al., 2020a*. This meta-analysis also focussed on individuals with European ancestries and derived data from the IAMDGC 2013 (17,181 cases and 60,074 controls) (*Fritsche et al., 2013*) and IAMDGC 2016 (16,144 advanced AMD cases and 17,832 controls) (*Fritsche et al., 2016*) studies as well as the GERA study (4017 cases and 14,984 controls) (*Kvale et al., 2015*). The relevant summary statistics are primarily reflective of advanced AMD, but the GERA cohort included both advanced and intermediate AMD cases. Advanced AMD was broadly defined by the presence of geographic atrophy or choroidal neovascularisation, although there was a degree of variability in the criteria used in the included studies. Notably, the MTAG approach can leverage the high genetic correlation between the input phenotypes to detect genetic associations relevant only to advanced AMD.

#### Exposure data

A phenome-wide screen was performed to make causal inferences on the role of a wide range of traits in early and advanced AMD. To achieve this, both published and unpublished GWAS data from the IEU open GWAS database were used; these were accessible via the TwoSampleMR programme in R (*Hemani et al., 2018*). All European GWAS within this database were included with the exception of imaging phenotypes and expression quantitative trait locus related data which were removed. The restriction to European datasets limits the generalisability of the results to other populations but is necessary to produce reliable findings. In the early AMD analysis, studies from the 'ukb' and 'met-d' batches were excluded as data for these studies were entirely from the UK Biobank resource and, as a result, there was extensive population overlap with the early AMD GWAS (*Sudlow et al., 2015*). In the advanced AMD analysis, the 'ukb' and 'met-d' batches were included. The early AMD analysis was conducted on 30/12/2021 and a total of 10,979 traits were considered for analysis. The advanced AMD analysis was conducted on 08/01/2022. On 26/01/2022 we added the newly published 'finn-b' (*n* = 2803) traits to the analysis in place of the outdated 'finn-a' traits (*n* = 1489). It was impractical to manually inspect the degree of population overlap for all traits prior to conducting the analysis; instead, the degree of overlap for all significant traits was inspected after the analysis.

### Instrument selection

A statistically driven approach to instrumental variable selection was used. Typically, an arbitrary p-value threshold is set for the identification of appropriate single-nucleotide variants (SNVs); these are subsequently used as instrumental variables (referred to thereafter as *instruments*). A conventional p-value threshold for the selection of instruments is >5E−08. This approach however can, in some cases, be problematic. For example, when the number of instruments exceeding this threshold is small, the analysis can be underpowered or, in certain cases of unbiased screens, the results can be inflated (*Boddy et al., 2022*). With this in mind, the p-value for instrument selection for each trait was set to the level where >5 instruments were available for each analysis. More specifically, for each trait, the analysis would first be conducted with a p-value threshold for inclusion of 5E−8 before sequentially increasing the threshold by a factor of 10 each time until >5 eligible instruments are identified.

A predefined maximum p-value of 5E−05 was used and the final range of pvalues for inclusion was 5E−06 to 5E−08.

## Proxies

Where an exposure instrument was not present in the outcome dataset, a suitable proxy was identified (*Hartwig et al., 2016*). In the early AMD analysis, this was achieved by using the TwoSampleMR software with a linkage disequilibrium $R^2$ value of ≥0.9 (*Purcell et al., 2007*). For the advanced AMD phenotype, data that were not derived from the TwoSampleMR resource were used and therefore the Ensembl server was utilised to identify proxies (*Cunningham et al., 2022*; *Hemani et al., 2018*).

## Clumping

SNVs were clumped using a linkage disequilibrium $R^2$ value of 0.001 and a genetic distance cut-off of 10,000 kilo-bases. A European reference panel was used for clumping.

## Harmonisation

The effects of instruments on outcomes and exposures were harmonised to ensure that the beta values (i.e. the regression analysis estimates of effect size) were expressed per additional copy of the same allele (*Hartwig et al., 2016*). Palindromic alleles (i.e. alleles that are the same on the forward as on the reverse strand) with a minor allele frequency >0.42 were omitted from the analysis in order to reduce the risk of errors.

## Removal of pleiotropic genetic variants and outliers

Pleiotropic instruments and outliers were removed from the analysis by using a statistical approach that removes instruments which are found to be more significant for the outcome than for the exposure (*Hemani et al., 2017*). Radial MR, a simulation-based approach that detects outlying instruments, was also utilised (*Bowden et al., 2018*).

## Causal inference

MR relies on three key assumptions with regard to the instrumental variable: (1) the instrumental SNV should be associated with the exposure; (2) the SNV should not be associated with confounders; (3) the SNV should influence the outcome only through the exposure (*Julian et al., 2021*).

MR estimation was primarily performed using a multiplicative random effects (MRE) inverse variance weighted (IVW) method. MRE IVW was selected over a fixed effects (FE) approach as it allows inclusion of heterogeneous instruments (this was certain to occur within the breadth of this screen) (*Burgess et al., 2019*). A range of 'robust measures' were used to increase the accuracy of the results and to account for violations of the above key MR assumptions (*Burgess et al., 2019*); these measures included weighted median (*Bowden et al., 2016a*), Egger (*Burgess and Thompson, 2017*), weighted mode (*Hartwig et al., 2017*), and radial MR with modified second-order weights (*Bowden et al., 2018*).

## Further quality control

The instrument strength was determined using the *F*-statistic (which tests the association between the instruments and the exposure) (*Burgess and Thompson, 2011*). *F*-Statistics were calculated against the final set of instruments that were included. A mean *F*-statistic >10 was considered sufficiently strong.

The Cochran's *Q* test was performed for each analysis. Cochran's *Q* is a measure of heterogeneity among causal estimates and serves as an indicator of the presence of horizontal pleiotropy (which occurs when an instrument exhibits effects on the outcome through pathways other than the exposure) (*Bowden and Holmes, 2019*). It is noted that a heterogeneous instrument is not necessarily invalid, but rather calls for a primary assessment with an MRE IVW rather than an FE approach; this has been conducted as standard throughout our analysis.

The MR-Egger intercept test was used to detect horizontal pleiotropy. When this occurs, the Egger regression is robust to horizontal pleiotropy (under the assumption that that pleiotropy is uncorrelated with the association between the SNV and the exposure) (*Burgess and Thompson, 2017*). Unless otherwise indicated by the Egger intercept, the assumption that no demonstrable horizontal

pleiotropy is present was made and Egger regression was not utilised to determine causal effects (given the low power of this approach in the context of a small number of SNVs) (*Bowden et al., 2015*).

The $I^2$ statistic was calculated as a measure of heterogeneity between variant-specific causal estimates. An $I^2 < 0.9$ indicates that Egger is more likely to be biased towards the null through violation of the 'NO Measurement Error' (NOME) assumption (*Bowden et al., 2016b*).

Leave-one-out cross-validation was performed for every analysis to determine if any particular SNV was driving the significance of the causal estimates.

## Management of duplicate traits

As the GWAS database that was used contained multiple different GWAS for certain traits, some exposures were analysed on multiple occasions. Where this occurred, the largest sample size study was considered to be the primary analysis. Where there were duplicate studies in the same population, the study with the largest *F*-statistic was used.

## Identification of significant results

Before considering an MR result to be significant, the results of a range of causal inference and quality control tests should be taken into account. Notably, it is not necessary for a study to find significance in all measures to determine a true causal relationship. MR is a low power study type and, as such, an overly conservative approach to multiple testing can be excessive (*Burgess et al., 2019*). However, in the context of the present study the results of the early AMD phenome-wide screen were considered significant only if they remained: significant after false discovery rate (FDR) correction in the MRE IVW; nominally significant in weighted mode and weighted median; and nominally significant throughout the leave-one-out analysis (MRE IVW) (*Benjamini and Hochberg, 1995*). This conservative approach was selected as a large number of phenotypes was studied and because we wanted to focus on high confidence signals.

Where causal traits for early AMD were identified, the relationship between these traits and advanced AMD was studied. These two AMD classifications are phenotypically distinct but are generally part of the same disease spectrum. When traits failed to replicate as causal factors in the advanced AMD dataset, it could not be inferred that these traits are not truly causal for early AMD. However, significance in both AMD phenotypes provided support for the detected causal links and evidence that a factor plays a role across the disease spectrum.

## Multivariable MR

Multivariable MR was performed in circumstances where it was important to estimate the effect of >1 closely related (and/or potentially confounding) exposure trait (*Sanderson et al., 2019*). P-values for the inclusion of instruments for the exposures of interest were optimised to obtain sufficiently high (>10) conditional *F*-statistics for reliable analysis (*Sanderson et al., 2021*). With this in mind, selection for exposures began at a p-value threshold of >5E−08. Where trait's instruments had a conditional *F*-statistic <10, the p-value for selection was reduced in an automated manner by factor of 10 until an *F*-statistic >10 was obtained. The same clumping procedure as in the univariable MR analysis was used. Adjusted Cochran's *Q*-statistics were calculated, with a p-value of <0.05 indicating significant heterogeneity. Where the Cochran's *Q*-statistic indicated heterogeneity, a *Q*-statistic minimisation procedure was used to evaluate the causal relationship; testing assumed both high (0.9) and low (0.1) levels of phenotypic correlation (*Sanderson et al., 2021*).

## Two-sample multivariable Mendelian randomisation approach based on Bayesian model averaging (MR-BMA)

Multivariable MR can be used to obtain effect estimates for a few (potentially related) traits. However, it cannot be directly applied when many traits need to be considered. In contrast, Mendelian randomisation Bayesian model averaging (MR-BMA), a Bayesian approach first described by *Zuber et al., 2020*, can search over large sets of potential risk factors to determine which are most likely to be causal.

Notably, *Zuber et al., 2020* previously performed an in-depth analysis which considered the role of lipids against an older AMD GWAS. The relevant study served as proof-of-concept for the MR-BMA

method and demonstrated that several lipid traits have causal roles in AMD. However, the analysis had two potential limitations. First, it downweighed fatty acid traits through limiting composite traits for SNV identification to HDL, LDL, and triglycerides. Second, numerous lipid traits with a potential role in AMD were not included in the analysis. For these reasons, we chose to conduct a more comprehensive analysis with a slightly altered approach. The following study design modifications were made compared to the study by *Zuber et al., 2020*:

1. Fatty acids were included as a composite trait (utilising GWAS data for serum fatty acids derived from the Nightingale Health 2020 resource as listed in TwoSampleMR package [*Hemani et al., 2018*]).
2. All lipid and fatty acid measures included in a GWAS by *Kettunen et al., 2016* were considered as potential causal traits ($n$ = 102 traits).
3. A more recent AMD GWAS was used (*Winkler et al., 2020*).
4. In general, multivariable MR (of any sort) cannot produce reliable results where the studied traits are ≥0.99 correlated with respect to the included instruments. For this reason, where two traits were highly correlated, one was removed at random rather than by manually selecting traits in a manner which risks selection bias.

MR-BMA for immune cell and complement phenotypes was additionally performed. In this analysis, instruments were obtained at genome-wide significance (p-value >5E−08) for every included exposure in the model (given that composite traits were not available/applicable). For the immune cell MR-BMA, all immune traits that were studied in a GWAS by *Orrù et al., 2020* and were present in the TwoSampleMR package were used as exposures. In the complement analysis, all complement traits available in relevant studies by *Sun et al., 2018* and *Suhre et al., 2017* were used, with the exception of complement subfractions.

For the MR-BMA analysis, the prior probability was set to 0.1 and the prior variance was set to 0.25. A stochastic search with 10,000 iterations was undertaken and empirical pvalues with 100,000 permutations were calculated.

## Presentation of effect sizes

Effect sizes are presented to enable appraisal of the impact of putative risk factors. For the univariable MR analysis, these are presented as OR per 1 standard deviation of change in the exposure for continuous traits, and as beta values for binary exposure traits. This approach was selected because ORs are uninformative for binary exposure traits (*Burgess and Labrecque, 2018*). Furthermore, beta values are not presented by all MR authorities where an exposure variable is binary and it is often highlighted that these values are only indicative. Multivariable MR effect sizes are presented as beta value estimates irrespective of the nature of the exposure variable (given that the role of multivariable MR within this study was purely to identify confounders).

MR-BMA effect sizes are presented in the form of a model-averaged causal estimate (MACE). The MACE is a conservative estimate of the direct causal effect of an exposure on an outcome averaged across models. It is noted that the primary function of MR-BMA is to highlight the probable causal trait among a number of candidate causal risk factors. Although the MR-BMA findings can be used to interpret the direction of effect, they should not be necessarily viewed as an absolute guide to magnitude (*Zuber et al., 2020*).

## Software

R version 4.1.0.
TwoSampleMR version 0.5.6.
RadialMR version 1.0.
MVMR version 0.3.
MR-BMA code was sourced from https://github.com/verena-zuber/demo_AMD (*Zuber, 2021*; *Zuber et al., 2020*).

## Results

### Overview

Focussing on early AMD, univariable MR analysis was applied to a broad range of traits. Following quality control, significant results were replicated in an advanced AMD dataset and further analyses were conducted using multivariable MR and MR-BMA (*Zuber et al., 2020*).

Overall, 4591 traits were eligible for analysis. Among these, 44 were found to be putatively causal for early AMD (*Table 1*, *Source data 1*). Most of these causal traits were serum lipoprotein concentration and compositional measures (*n* = 29). Other significant traits identified included immune cell phenotypes (*n* = 5), serum proteins (*n* = 6), and disease phenotypes (*n* = 4).

### Lipoprotein metabolism is linked to AMD risk

Univariable MR demonstrated significant causal relationships for 28 serum lipoprotein measures and 1 serum fatty acid concentration (18:2 linoleic acid) in early AMD (*Table 1*). These relationships were also strongly supported by the results of our advanced AMD analysis (*Source data 1*).

Serum metabolites are highly correlated traits, and the instruments for serum lipoprotein and fatty acid measures in the present study were demonstrably correlated (*Figure 1*). MR-BMA was used in order to discern which traits were driving the causal relationships. In our initial analysis, two genetic variants (rs11065987 [*BRAP*] and rs10455872 [*LPAL2*]) were found to be outliers in terms of *Q*-statistic (*Q*-statistic 13.84–15.71 and 6.79–10.46, respectively, across models) (*Figure 2*). These SNVs were therefore omitted and the analysis was re-run. In subsequent analyses, no outlier SNVs were identified. The top 10 models in terms of posterior probability are presented in the *Source data 1*; the top 10 individual risk factors in terms of marginal inclusion probability (MIP; defined as the sum of the posterior probabilities over all the models where the risk factor is present) are shown in *Table 2*. The MIPs of all the traits included this analysis are plotted in *Figure 3*. The top 4 traits with respect to their MIP were serum sphingomyelins (MIP = 0.76, MACE = 0.29), triglycerides in IDL (MIP = 0.32, MACE = −0.16), free cholesterol (MIP = 0.20, MACE = 0.07), and phospholipids in very small VLDL (MIP = 0.63, MACE = −0.31). It is noted that whilst MR-BMA is designed to select the likely causal risk factor among a set of candidate causal traits, it is often not possible to achieve this with certainty. As such, a definitive statement of causality for individual lipid traits cannot be obtained within the utilised MR causal inference framework.

Serine palmitoyltransferase 2, an enzyme which catalyses the first committed step in sphingolipid biosynthesis, was robustly associated with early AMD. The detected effect size highlighted that genetic liability to increasing serum enzyme levels is protective of AMD (*Table 1*,*Source data 1*; OR 0.86, p-value = 1.00E−03). It is of interest that increasing serum levels of the related enzyme serine palmitoyltransferase 1 also appeared to be protective of early AMD in most measures (OR 0.94, IVW p-value = 2.00E−03, FDR-adjusted IVW p-value = 0.05, weighted median p-value = 0.05; significant throughout leave-one-out analysis). Serine palmitoyltransferase enzymes were not significantly related to the risk of advanced AMD (*Source data 1*).

### Reinforcing a causal role for complement

Increasing serum levels of the complement proteins CD59 glycoprotein (OR 1.10, IVW p-value = 2.04E−05) and complement factor H-related protein 5 (OR 1.10, IVW p-value = 7.70E−07) were found to be associated with genetic liability to early AMD and passed multiple testing correction. Whilst complement C4, complement factor B and complement factor I were also related to early AMD risk at nominal level of significance, they did not exceed the conservative criteria for causal inference imposed in this study with respect to robust measures (*Source data 1*). In subsequent advanced AMD analyses (*Source data 1*), CD59 glycoprotein remained related to disease risk (OR 1.07, IVW p-value = 0.04, weighted median p-value = 9.00E−04, MR-Egger p-value = 2.00E−03, weighted mode p-value = 7.00E−03). Complement factor I reached a nominal level of significance in the IVW measure (OR 0.95, p-value = 0.05) but did not remain significant in other tests. No other complement measures that were nominally significant for early AMD were found to be significantly related to advanced AMD.

Complement traits are correlated with one another, and as such MR-BMA analysis was performed. In the initial analysis, the relationship between complement proteins was strongly suggested to be driven by complement factor H (posterior probability = 0.83, MIP = 1.00, MACE = −0.68). However, rs2274700 (*CFHR3*, Cook's distance 26.26–55.24 and Cochran's *Q* 0.87–2.12 across models) and

Table 1. Significant results detected in a phenome-wide univariable Mendelian randomisation (MR) analysis of early age-related macular degeneration (AMD).
Only traits passing the conservative quality control criteria described in the methods are listed.

| Trait name | Odds ratio | FDR-adjusted IVW p value | MRE IVW beta |
|---|---|---|---|
| Rheumatoid arthritis | NA | 1.51E−06 | 0.08 |
| Unswitched memory B cell % B cell | 1.11 | 5.40E−05 | 0.10 |
| CD62L− dendritic cell % dendritic cell | 0.95 | 3.32E−02 | −0.05 |
| Effector memory CD8+ T cell absolute count | 1.08 | 1.79E−04 | 0.07 |
| CD25 on IgD+CD38− naive B cell | 0.93 | 1.25E−04 | −0.07 |
| CD80 on plasmacytoid dendritic cell | 0.96 | 1.63E−02 | −0.04 |
| Total cholesterol in IDL* | 0.80 | 6.48E−08 | −0.22 |
| Free cholesterol in IDL* | 0.80 | 5.51E−08 | −0.22 |
| Total lipids in IDL* | 0.79 | 3.35E−09 | −0.23 |
| Concentration of IDL particles* | 0.80 | 9.83E−09 | −0.23 |
| Phospholipids in IDL* | 0.79 | 1.86E−09 | −0.23 |
| Triglycerides in IDL* | 0.84 | 2.40E−07 | −0.18 |
| Total cholesterol in large LDL* | 0.83 | 9.85E−08 | −0.19 |
| Cholesterol esters in large VLDL* | 0.83 | 1.41E−07 | −0.19 |
| Free cholesterol in large LDL* | 0.83 | 7.07E−07 | −0.18 |
| Total lipids in large LDL* | 0.83 | 1.04E−07 | −0.19 |
| Concentration of large LDL particles* | 0.83 | 2.40E−07 | −0.19 |
| Phospholipids in large LDL* | 0.83 | 1.33E−06 | −0.18 |
| Cholesterol esters in large VLDL* | 0.82 | 6.55E−03 | −0.20 |
| 18:2 linoleic acid (LA)* | 0.80 | 2.93E−07 | −0.23 |
| Total cholesterol in LDL* | 0.82 | 4.46E−08 | −0.19 |
| Total cholesterol in medium LDL* | 0.82 | 3.64E−09 | −0.20 |
| Cholesterol esters in medium LDL* | 0.82 | 5.93E−09 | −0.20 |
| Total lipids in medium LDL* | 0.81 | 3.35E−09 | −0.21 |
| Concentration of medium LDL particles* | 0.81 | 3.64E−09 | −0.21 |
| Phospholipids in medium LDL* | 0.81 | 1.86E−09 | −0.21 |
| Total cholesterol in small LDL* | 0.81 | 9.85E−08 | −0.21 |
| Total lipids in small LDL* | 0.81 | 2.65E−07 | −0.21 |
| Total cholesterol in small VLDL* | 0.85 | 2.54E−02 | −0.16 |
| Serum total cholesterol* | 0.77 | 3.67E−08 | −0.26 |
| Total phosphoglycerides* | 0.81 | 3.93E−03 | −0.21 |
| Triglycerides in very large HDL* | 0.87 | 4.37E−04 | −0.14 |
| Total lipids in very small VLDL* | 0.84 | 1.97E−03 | −0.18 |
| Concentration of very small VLDL particles* | 0.84 | 9.04E−04 | −0.18 |
| Phospholipids in very small VLDL* | 0.83 | 3.63E−03 | −0.19 |
| Interferon alpha-10 | 1.14 | 1.83E−02 | 0.13 |
| Protein S100-A5 | 1.07 | 6.94E−04 | 0.07 |

*Table 1 continued on next page*

*Table 1 continued*

| Trait name | Odds ratio | FDR-adjusted IVW p value | MRE IVW beta |
|---|---|---|---|
| Serine palmitoyltransferase 2 | 0.86 | 3.17E−02 | −0.15 |
| CD59 glycoprotein | 1.10 | 1.77E−03 | 0.09 |
| Complement factor H-related protein 5 | 1.09 | 9.30E−05 | 0.09 |
| Cathepsin F | 1.10 | 4.44E−03 | 0.10 |
| Benign neoplasm: skin, unspecified | NA | 8.23E−03 | 0.07 |
| Psychiatric diseases | NA | 8.54E−05 | −0.43 |
| Myotonic disorders | NA | 2.32E−02 | 0.00 |

FDR, false discovery rate; IVW, inverse variance weighted; MRE, multiplicative random effects; %, as a proportion of; NA, not applicable (as exposure traits of binary nature do not produce accurate odds ratios; beta values can be used instead to infer direction of effect but not necessarily magnitude).
*These traits (trait ID group 'met-c' in **Source data 1**) had 1.9% sample overlap with the early AMD dataset, a minor degree of overlap which is unlikely to bias results; the remaining causal traits had no sample overlap with early AMD.

rs10824796 (*MBL2*, Cook's distance 0.13–9.93 and Cochran's *Q* 0.87–22.12 across models) were identified as outlier instruments and therefore warranted removal from subsequent analyses. Rs2274700, a genetic variant known to be associated with AMD risk (*Liao et al., 2016*), represented the only instrument strongly associated with complement factor H in the MR-BMA analysis and, as such, its removal precluded an informative high-throughput analysis of the complement cascade. Whilst it is clear that there is a causal relationship between complement and AMD, it is not possible to comment with precision about the specific complement-related molecule driving this.

## Other immune traits have mixed protective and causal effects in AMD

Our screen identified other serum immune traits with potential causal roles in early AMD, with several of these related to dendritic cell populations. These traits were: Unswitched memory B cell as a proportion of B cells (OR = 1.11, p-value = 4E−07); CD62L− dendritic cell as a proportion of dendritic cells (OR = 0.95, p-value = 1.22E−03); Effector memory CD8+ T cell absolute count (OR = 0.95, p-value = 1.56E−06); CD80 on plasmacytoid dendritic cell (OR = 0.96, p-value = 4.20E−04); and Interferon alpha 10 (OR = 1.14, p-value = 5.38E−04). Notably, numerous immune cell traits were also found to be significantly associated with advanced AMD (*Source data 1* ). As with lipids, these traits are highly correlated and it is not possible to confidently pinpoint which immune cell trait is truly causal in a univariable analysis due to potential overlap in terms of genetic instruments. MR-BMA (using data from a GWAS by *Orrù et al., 2020*) for the above immune traits was not possible due to the presence of too few genetic variants strongly instrumenting across the exposure variables which, in turn, led to a failure to construct a meaningful model. For this reason, it was not possible to determine which of these specific factors are driving the casual relationship between immune cell traits and AMD (though dendritic cell traits dominate the univariable analysis).

## Disease phenotypes are related to AMD risk

There are four disease phenotypes/groups which were found to be significant at a level which satisfy the criteria for causal inference defined in this study. These traits are rheumatoid arthritis (beta = 0.08, IVW p-value = 9.86E−09), psychiatric diseases (beta = −0.43, IVW p-value = 6.88E−07), 'benign neoplasm: skin, unspecified' (beta = 0.07, IVW p-value = 1.76E−04), and myotonic disorders (beta = −0.003, IVW p-value = 7.49E−04). None of these disorders were significant in advanced AMD. The very broad and non-specific definitions encapsulated in the GWAS of benign skin neoplasms, psychiatric disease, and myotonic disorder make it challenging to apply further analyses within the scope of this study, but these may be interesting phenotypes for exploration in future studies.

In univariable MR, rheumatoid arthritis was identified to be causally related to AMD as detailed above. Multivariable MR with effector memory CD8+ T cell absolute count as a second exposure

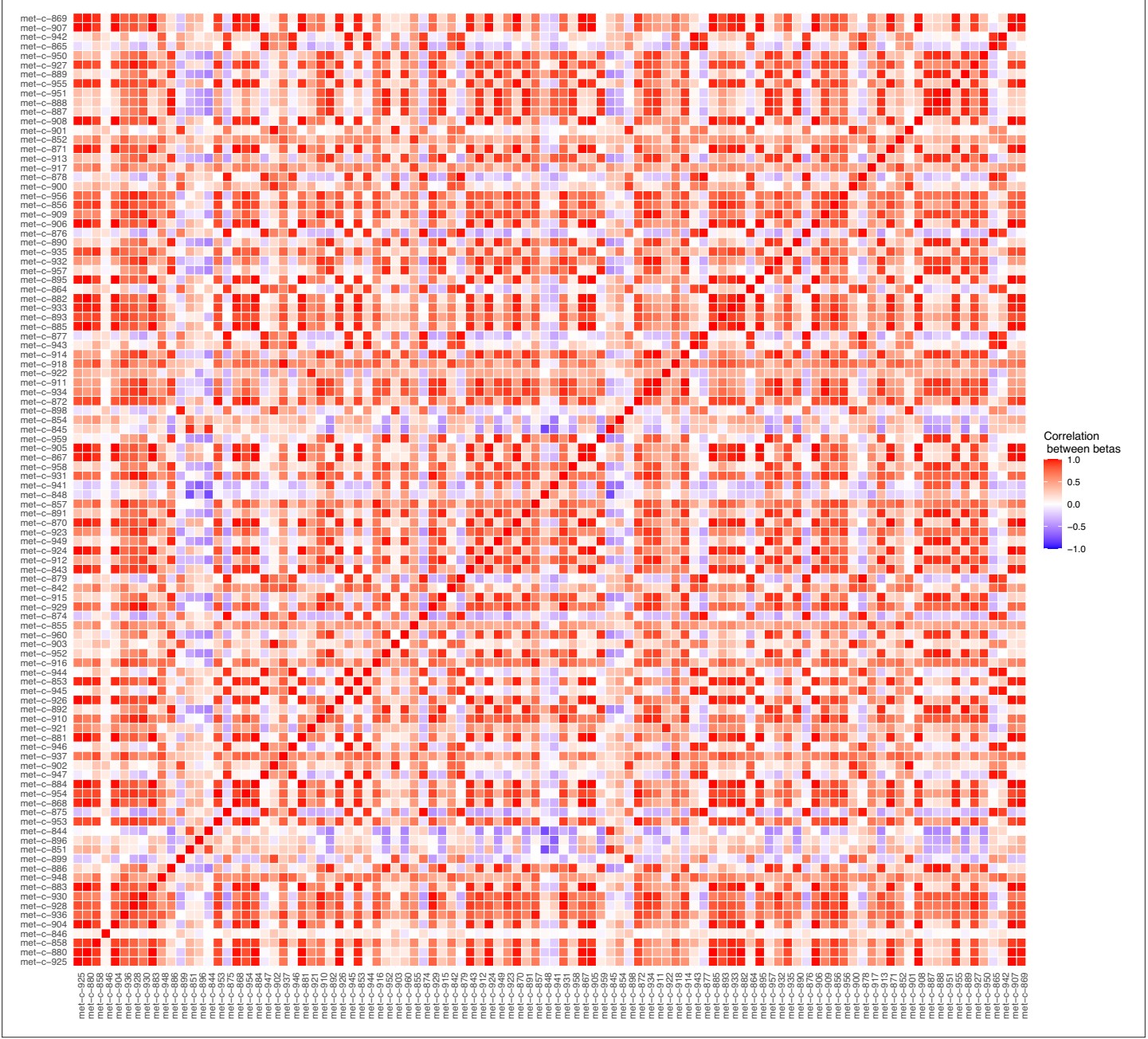

**Figure 1.** Plot illustrating the correlations between the beta values for the metabolites considered in our Mendelian randomisation Bayesian model averaging (MR-BMA) analysis for early age-related macular degeneration (AMD). This plot visually represents the correlation matrix between the genetic associations of the exposure variables with respect to their instruments. The traits are labelled according to their 'Trait ID'; further information can be found in the *Source data 1*.

variable (utilising a *Q*-statistic minimisation procedure) demonstrated that the causal effect of rheumatoid arthritis was either mediated by immune cells or underpinned by correlated pleiotropy (rheumatoid arthritis effect size: 0.03, 95% CI: −0.007 to 0.055; effector memory CD8+ T cell absolute count effect size 0.07, 95% CI: 0.001 to 0.19). This finding persists in models assuming both very high (0.9) and low (0.1) phenotypic correlation.

With respect to the relationship between AMD and psychiatric disease, no reverse causation was identified (p = 0.31). More specific psychiatric traits which could be analysed were not significantly related to genetic liability to early AMD (*Source data 1*). An extensive range of psychiatric disorders were also explored in the advanced AMD dataset but no significant relationships were detected

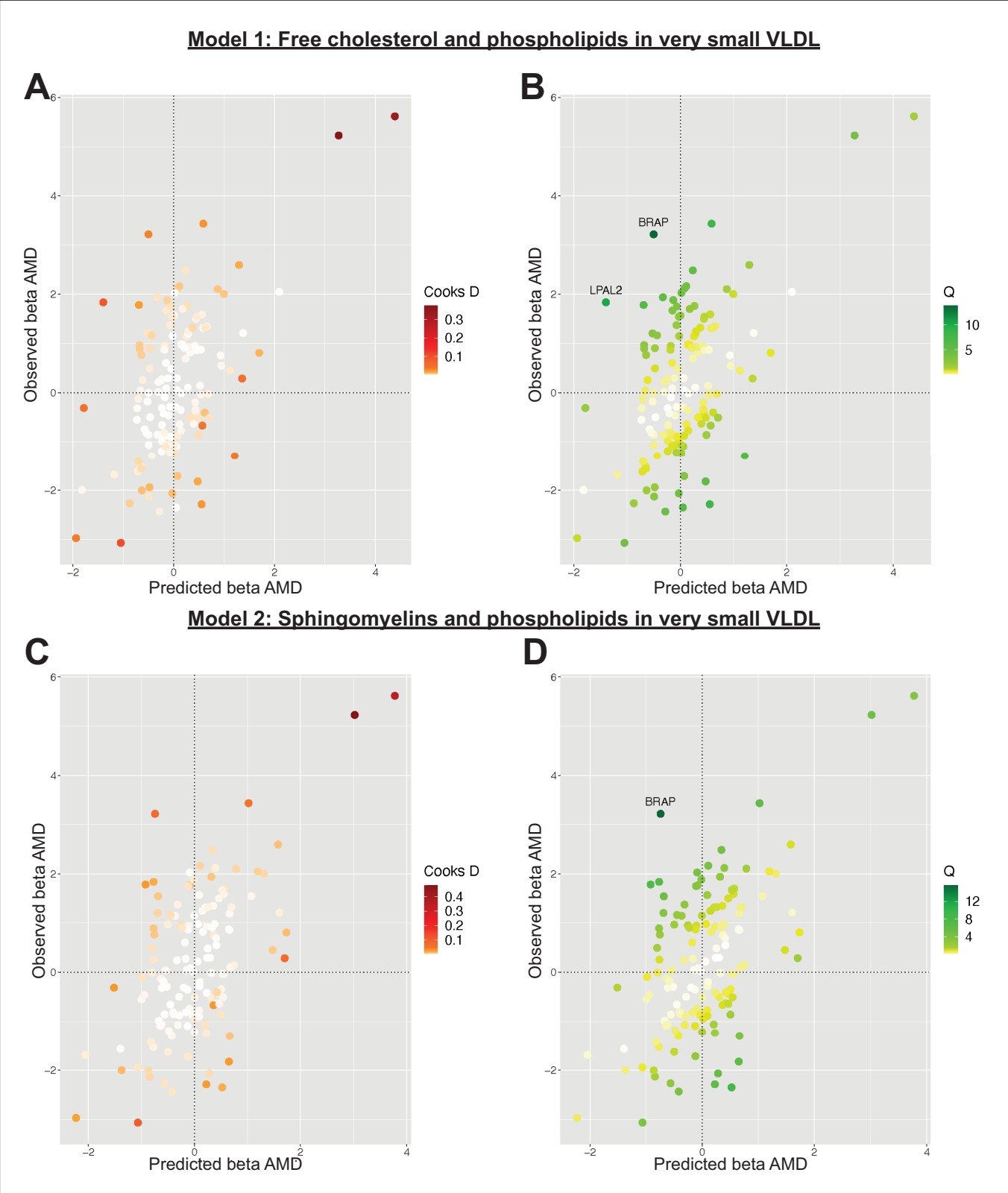

**Figure 2.** Plots outlining the top-ranking models with respect to their posterior probability in the first run of Mendelian randomisation Bayesian model averaging (MR-BMA) of lipid-related traits in early age-related macular degeneration (AMD). Plots (A) and (C) present Cook's distance while plots (B) and (D) present Cochran's Q. Outlier instruments are annotated. The Cochran's Q is a measure which serves to identify outlier variants with respect to the fit of the linear model. The Q-statistic is used to identify heterogeneity in a meta-analysis, and to pinpoint specific variants as outliers. The contribution

*Figure 2 continued on next page*

*Figure 2 continued*

of variants to the overall *Q*-statistic is measured (defined as the weighted squared difference between the observed and predicted association with the outcome) in order to identify outliers. Cook's distance on the other hand is utilised to identify influential observations (i.e. those variants which have a strong association with the outcome). Such variants are removed from the analysis because they may have an undue influence over variable selection, leading to models which fit that variant well but others poorly.

(*Source data 1*). Due to the broad nature of this trait, and the negative results for all disorders explored, it was not possible to discern the driving signal behind the significant result for psychiatric disease. It appears probable that psychiatric disease is a false-positive result given that it lacks: the supporting evidence of biological plausibility; validation in an advanced AMD dataset; or supportive evidence through significance of other similar traits in the early AMD analysis.

### Serum cathepsin F and S100 proteins have a causal role in AMD

In addition to the four previously mentioned molecules (*Table 1*: Interferon alpha-10, CD59 glycoprotein, complement factor H-related protein and serine palmitoyltransferase 2) a further two serum proteins were identified to have likely causal effects. These are S100-A5 (OR 1.07, IVW p-value = 6.80E−06) and cathepsin F (OR = 1.10, IVW p-value = 7.16E−05).

Serum cathepsin F was shown to be causally related to early AMD. However, no other molecules from the serum cathepsin group produced significant results for early AMD and no cathepsin group protein was robustly significant for advanced AMD (although cathepsins S and G have significant IVW p-values of 1.71E−05 and 0.001, respectively; *Source data 1*).

Serum protein S100-A5 was demonstrated to be causal for AMD (OR 1.07, IVW p-value = 6.80E−06). Additionally, protein S100-A13 was significantly related to early AMD risk in both IVW (OR = 1.07, p-value = 1.22E−05) and other measures (weighted median p-value = 3.31E−05, weighted mode p-value = 9.00E−04, Egger p-value = 1.00E−03) but did not remain significant throughout the leave-one-out analysis, suggesting that the relationship for this particular protein could be driven by a small number of influential variants. Whilst protein S100-A2 was nominally significant in the IVW analysis (p = 0.04), it was not found to be significant in other measures. No S100 group proteins were significantly related to advanced AMD risk (*Source data 1*).

## Discussion

In this study, we used MR to advance understanding of AMD pathogenesis. Our findings indicate protective effects for serum VLDL and IDL compositional and concentration traits in AMD. Causal

**Table 2.** Lead causal traits identified by Mendelian randomisation Bayesian model averaging (MR-BMA) of lipid-related phenotypes in early age-related macular degeneration (AMD) ranked according to their marginal inclusion probability (MIP).

| Rank | Risk factor (trait ID) | MIP | Average effect | Nominal p-value | FDR-adjusted p-value |
|---|---|---|---|---|---|
| 1 | Sphingomyelins (met-c-935) | 0.76 | 0.30 | 2.40E−04 | 5.76E−03 |
| 2 | Phospholipids in very small VLDL (met-c-955) | 0.63 | −0.31 | 1.00E−05 | 7.20E−04 |
| 3 | Triglycerides in IDL (met-c-872) | 0.32 | −0.16 | 2.10E−04 | 5.76E−03 |
| 4 | Free cholesterol (met-c-858) | 0.20 | 0.07 | 2.83E−03 | 5.09E−02 |
| 5 | Omega-3 fatty acids (met-c-855) | 0.07 | 0.01 | 3.70E−02 | 2.34E−01 |
| 6 | Free cholesterol in very large HDL (met-c-944) | 0.07 | 0.01 | 2.74E−02 | 2.34E−01 |
| 7 | Total lipids in very small VLDL (met-c-953) | 0.06 | −0.02 | 2.08E−02 | 2.34E−01 |
| 8 | Cholesterol esters in medium VLDL (met-c-910) | 0.05 | 0.01 | 3.66E−02 | 2.34E−01 |
| 9 | Cholesterol esters in very large HDL (met-c-943) | 0.05 | 0.01 | 4.87E−02 | 2.34E−01 |
| 10 | Ratio of bisallylic groups to double bonds (met-c-844) | 0.05 | 0.01 | 4.41E−02 | 2.34E−01 |

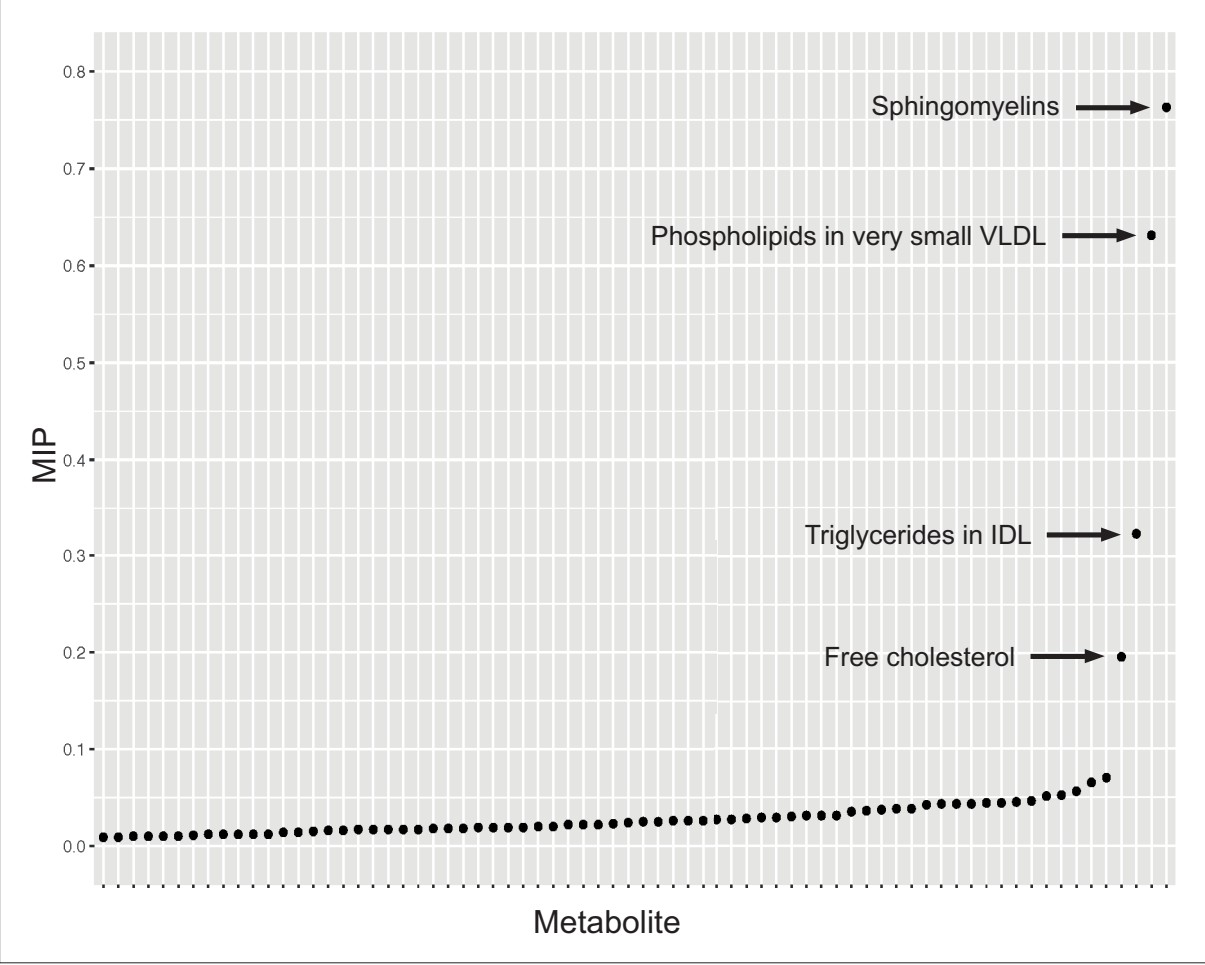

**Figure 3.** Graph detailing the results of a Mendelian randomisation Bayesian model averaging (MR-BMA) analysis that aims to identify causal lipid-related risk factors for early age-related macular degeneration (AMD). The studied phenotypes are ranked according to their marginal inclusion probability (MIP); four likely causal traits are highlighted.

effects for serum-free cholesterol and sphingolipid metabolism were also highlighted. Notably, our results suggest that increasing serum levels of sphingomyelins are causally linked to AMD, and we report protective relationships for specific enzymes implicated in sphingolipid biosynthesis. It is known that sphingolipids play key roles in retinal physiology, and there is emerging evidence that some sphingolipids may play a role in AMD pathogenesis (*Simon et al., 2021*). Further, our findings are in keeping with those of a meta-analysis of metabolomic studies in which pathway enrichment analysis suggested a role for sphingolipid metabolism in AMD (*Hou et al., 2020*). Whilst sphingomyelin itself has not previously been described as a causative factor for AMD, other substances involved in sphingolipid metabolism (including ceramide, which can serve as both a substrate for sphingomyelin synthesis or a product of sphingomyelin metabolism) have been suggested as causal factors through regulation of retinal cell death, inflammation, and neovascularisation (*Simon et al., 2021*). It is known that the balance of sphingolipids has crucial roles in determining cell fates, and, therefore, the contrasting effects of serine palmitoyltransferase and sphingomyelin observed in this study are noteworthy (*Taniguchi and Okazaki, 2020*). Our study therefore adds weight to the assertion that sphingolipids play a key role in AMD pathophysiology. Further study of sphingolipid metabolism in individuals with AMD is expected to provide important insights and to highlight potential treatment targets.

Our MR findings support the well-documented assertion that complement plays a key role in AMD pathogenesis (*Clark and Bishop, 2018*; *Jha et al., 2007*; *Sivaprasad and Chong, 2006*). Whilst MR-BMA was not feasible for this set of traits, univariable analysis supported roles for complement

factor H-related protein 5 and CD59 glycoprotein. A causal signal was obtained for complement factor H-related protein 5, a finding in keeping with recent studies highlighting the role of this and other factor H-related proteins (FHR-1 to FHR4; all involved in the regulation of complement factor C3b turnover) in AMD (*Cipriani et al., 2021*). An intriguing observation was the apparently causal role of increasing serum levels of the complement-related CD59 glycoprotein (an inhibitor of membrane attack complex formation) in both early and advanced AMD. This finding appears to be at odds with previous studies that investigated the impact of high levels of CD59 in a model of AMD and found a potential therapeutic benefit (*Cashman et al., 2011*). The relationship between systemic and local complement activation is poorly understood and studies have produced conflicting results with respect to the roles of each complement molecule in AMD (*Clark and Bishop, 2018*; *Jha et al., 2007*). It can therefore be speculated that the superficially inconsistent results between our study and previous reports on CD59 reflect the complicated relationship between serum and retinal CD59 expression.

Notably, a connection between complement and lipid accumulation has been proposed in AMD. This is due to the observation that dysregulation of the complement system results in lipid deposition both systemically and in the retina. There is also metabolomic evidence that decreased VLDL and increased HDL levels are associated with increased complement activation independent of AMD status (*Armento et al., 2021*). Given our results, the relationship between VLDL and complement is an area warranting further study.

Cathepsins are a diverse family of lysosomal proteases which play an important part in cellular homeostasis by participating in antigen processing and by degrading chemokines and proteases (*Yadati et al., 2020*). Cathepsins are susceptible to age-related alterations and have been linked to modulation of pro-inflammatory signalling pathways. Additionally, cathepsins D and S play a direct role in the retina as they help with the degradation of photoreceptor outer segments. It has also been shown that homozygosity for variant B cystatin C (an inhibitor of cysteine proteases) causes reduced secretion of mature cystatin C and is associated with increased susceptibility to neovascular AMD (*Turk et al., 2012*; *Zurdel et al., 2002*). For these reasons, it has been suggested that dysregulation of cathepsin activity may be a factor in AMD pathophysiology. Here, we show data suggesting that serum cathepsin F, a ubiquitously expressed cysteine cathepsin, has a causal relationship with early AMD. The role of specific cathepsins in AMD is an area that has not yet been mechanistically explored.

S100 proteins act as intracellular regulators and extracellular signalling proteins. Intracellularly, S100 proteins have a wide range of roles including regulating proliferation, differentiation, apoptosis, calcium homeostasis, metabolism, and inflammation (*Donato et al., 2013*). Extracellular S100 proteins have important roles in immunity and inflammation. Of the S100 protein family, serum S100-A5 was the only protein to be robustly identified as a causal factor for AMD in our study. Whilst S100 proteins have received little study in AMD thus far, the binding of S100B to RAGE (Receptor for Advanced Glycation End products) and the subsequent increase in VEGF have previously been shown to be linked to the development of AMD (*Ma et al., 2007*; *Xia et al., 2017*). S100-A5 is a protein which has received little study and, as such, its biological functions are mostly uncharacterised (although is known to be involved in inflammation via the activation of RAGE) (*Wheeler and Harms, 2017*). Whilst it is outside the scope of this study to assert a mechanistic hypothesis for this finding, it seems likely that S100-A5 could be linked to inflammatory processes.

To a large extent, the present study has captured known AMD risk factors. A notable omission however is smoking, an accepted AMD contributor. In this study, cigarettes smoked per day was associated with early AMD at a nominal level of significance (IVW p-value = 0.009) but was not significant with FDR adjustment or in other measures. This finding is most probably reflective of the underpowered nature of MR. Given the hypothesis-free design of this study and the low power of MR, it would be inappropriate to conclude that non-significant results provide evidence of no relationship between certain environmental exposures and AMD.

In summary, this study has identified previously undescribed causal factors for AMD in addition to reinforcing several established AMD contributors. Future research will: explore how the interaction between causal traits affects AMD risk; investigate how the causal and protective factors described here alter the progression of established disease; follow-up these causal factors from a mechanistic perspective in order to open up avenues for therapeutic interventions.

## Acknowledgements

We acknowledge funding from the Wellcome Trust (224643/Z/21/Z; 200990/Z/16/Z); the University of Manchester's Wellcome Institutional Strategic Support Fund (Wellcome ISSF) grant (204796/Z/16/Z); the UK National Institute for Health Research (NIHR) Academic Clinical Fellow and Clinical Lecturer Programmes; Retina UK and Fight for Sight (GR586); the Australian National Health and Medical Research Council (NHMRC) (1150144).

## Additional information

### Competing interests

Stuart MacGregor: Cofounder of Seonix Bio Ltd; holds stock in Seonix Bio Ltd. Tariq Aslam: Involved in advisory boards and has received grants and speaker fees from Allergan, Novartis, Bayer, Roche, Bausch and Lomb, Heidelberg, Topcon and Canon; received consulting fees from Thea Pharmaceuticals. The other authors declare that no competing interests exist.

### Funding

| Funder | Grant reference number | Author |
|---|---|---|
| NIHR | Academic Clinical Fellow and Clinical Lecturer Programmes | Thomas H Julian Panagiotis I Sergouniotis |
| University of Manchester Wellcome ISSF for Causal Inference | 204796/Z/16/Z | Thomas H Julian Graeme CM Black Panagiotis I Sergouniotis Hui Guo Johnathan Cooper-Knock |
| Wellcome Trust Clinical Research Career Development Fellowship | 216596/Z/19/Z | Panagiotis I Sergouniotis |
| NHMRC | 1150144 | Stuart MacGregor |
| Fight for Sight and Retina UK | GR586 | Graeme CM Black |
| Wellcome Trust Transforming Genomic Medicine Initiative (TGMI) | 200990/Z/16/Z | Graeme CM Black |

The funders had no role in study design, data collection, and interpretation, or the decision to submit the work for publication. For the purpose of Open Access, the authors have applied a CC BY public copyright license [CC BY 4.0] to any Author Accepted Manuscript version arising from this submission.

### Author contributions

Thomas H Julian, Conceptualization, Data curation, Formal analysis, Funding acquisition, Investigation, Visualization, Methodology, Writing – original draft; Johnathan Cooper-Knock, Formal analysis, Supervision, Validation, Methodology, Writing – review and editing; Stuart MacGregor, Data curation, Methodology, Writing – review and editing; Hui Guo, Supervision, Methodology, Writing – review and editing; Tariq Aslam, Supervision, Writing – review and editing; Eleanor Sanderson, Supervision, Investigation, Methodology, Writing – review and editing; Graeme CM Black, Supervision, Project administration, Writing – review and editing, Funding acquisition; Panagiotis I Sergouniotis, Conceptualization, Supervision, Funding acquisition, Investigation, Methodology, Writing – original draft, Project administration, Writing – review and editing, Formal analysis

### Author ORCIDs

Thomas H Julian http://orcid.org/0000-0002-5488-5620
Panagiotis I Sergouniotis http://orcid.org/0000-0003-0986-4123

### Ethics

This study utilised publicly available data generated using human subjects; additional ethical approval was not required for this work.

### Decision letter and Author response

Decision letter https://doi.org/10.7554/eLife.82546.sa1
Author response https://doi.org/10.7554/eLife.82546.sa2

---

## Additional files

### Supplementary files

• MDAR checklist
• Source data 1. Source Data.

### Data availability

All data generated within this study are provided within the supplementary content. All genome-wide association study (GWAS) data utilised within this study are publicly available and can be sought in the original publications.

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
