## [Editor Report]

The findings of this study as well as the strength of the provided evidence are important and have significance beyond a single subfield. This manuscript is of interest to readers in the fields of ophthalmology, epidemiology and public health. The identification of both known and previously unknown risk factors for age-related macular degeneration (AMD) using genetically informed approaches can be combined with traditional epidemiological approaches to develop interventions that reduce the risk of AMD. The key claims of the manuscript are well supported by the data, and the approaches used are thoughtful and rigorous.

---

## [Decision Letter]

**Decision letter after peer review:**

Thank you for submitting your article "Phenome-wide Mendelian randomisation analysis identifies causal factors for age-related macular degeneration" for consideration by *eLife*. Your article has been reviewed by 2 peer reviewers, and the evaluation has been overseen by Lois Smith as the Reviewing and Senior Editor. The following individual involved in the review of your submission has agreed to reveal their identity: Eric Gaier (Reviewer #2).

*Reviewer #1 (Recommendations for the authors):*

Julian et al applied a Phenome-wide Mendelian randomization approach with the objective of identifying potential causal risk factors for age-related macular degeneration. Using 4,591 traits as potential exposures from an open access resource of GWAS summary statistics, the authors applied a suite of Mendelian randomization methods and sensitivity analyses to identify risk factors with robust causal association with early age-related macular degeneration. Follow-up analyses further investigated the causal associations with advanced age-related macular degeneration and applying MR Bayesian model averaging and multivariable MR to identify causal risk factors among closely related or highly correlated sets of traits. Of the 4,591 traits investigated, 44 were robustly associated with AMD - with protective effects identified for serum VLDL and IDL compositional and concentration traits; and causal effects also identified for serum-free cholesterol and sphingolipid metabolism. Overall, the manuscript is well written; the approaches used are thoughtful and rigorous; and the key claims of the manuscript are well supported by the data.

I only have a few comments.

1) What ancestry and reference panel were used for LD clumping.

2) Please provide the STOBE-MR checklist as a supplementary file.

3) The authors indicated that exposures containing the UKB were excluded to due to concerns regarding sample overlap and then for significant results, the degree of overlap with other cohorts was inspected after analysis. However, sample overlap for these analyses was not later reported. Was there any significant overlap that may have biased results?

*Reviewer #2 (Recommendations for the authors):*

Julian et al. set out to analyze the relative associations between exposure traits and AMD in an unbiased fashion. Their findings confirmed previously established associations, serving to validate their results, and also identified novel potentially causative traits. The results have wide-ranging impact in the AMD field.

Strengths:

1. Well-designed and executed study employing cutting-edge yet well-justified analytic approaches.

2. Clear rationale for approaches taken with real-time integration of established findings in AMD.

3. Conservative statistical interpretations guard against false-positive reporting and boosts relevance of the traits reported.

Weaknesses:

1. Limitations inherent to study design in which established databases are employed and clinical criteria and rigor by which they are applied for a specific disease entity are outside the investigators' control. These weaknesses are acknowledged, and the results are appropriately interpreted in these contexts.

1. Consider providing a supplemental label outlining the criteria for early and advanced AMD for the employed databases.

2. Consider expanding the discussion pertaining to potential interactions between traits found to be putatively causative. While individual traits may have been revealed as independently causative, how might interactions between these traits play into pathogenesis. Acknowledging that fully unpacking this is beyond the scope of this article, the discuss would benefit from the authors' overall impression and suggested future directions of experimentally assessing such potentially important interactions.

3. Consider expanding the discussion pertaining to traits found to be impactful in early but not advanced AMD. Might this reflect distinct causative factors within a subset of AMD prone to progression? Overall, the authors' don't seem to fully leverage the two datasets reflecting different stages of the same disease to make meaningful inferences about disease progression.

---

## [Author Response]

Reviewer #1 (Recommendations for the authors):1) What ancestry and reference panel were used for LD clumping.

We are grateful to the reviewer for noting our omission to describe the reference panel. We used a “European reference panel” (the default in TwoSampleMR and the most appropriate setting for the data that we used). We have now included a mention to this in the clumping section of the Methods (page 7, line 226).

2) Please provide the STOBE-MR checklist as a supplementary file.

We have now provided a STROBE-MR checklist as requested.

3) The authors indicated that exposures containing the UKB were excluded to due to concerns regarding sample overlap and then for significant results, the degree of overlap with other cohorts was inspected after analysis. However, sample overlap for these analyses was not later reported. Was there any significant overlap that may have biased results?

We are grateful to the reviewer for noting this omission. We manually inspected for sample overlap where an exposure was significantly related to early AMD after multiple testing correction. For ‘met-c’ traits, there was 1.9% sample overlap due to both exposure and outcome analyses using the ‘KORA S4’ cohort. We consider this degree of overlap to be low/negligible in terms of type 1 error risk. For all other statistically significant traits there was no sample overlap. We have now added a relevant statement in the results section (page 14, Table 1 legend). We have not commented on the sample overlap for traits that were not found to be statistically significant in our analysis as (i) this would require manual inspection of the manuscripts for the thousands of traits considered and (ii) sample overlap is generally associated with type 1 error, and as such, further dissection of non-significant traits is less crucial.

Reviewer #2 (Recommendations for the authors):1. Consider providing a supplemental label outlining the criteria for early and advanced AMD for the employed databases.

We thank the reviewer for this comment. We have now included a mention to the relevant criteria in Methods section (page 5, lines 135-151).

2. Consider expanding the discussion pertaining to potential interactions between traits found to be putatively causative. While individual traits may have been revealed as independently causative, how might interactions between these traits play into pathogenesis. Acknowledging that fully unpacking this is beyond the scope of this article, the discuss would benefit from the authors' overall impression and suggested future directions of experimentally assessing such potentially important interactions.

We fully agree with the above comment about the importance of interactions. As mentioned, the MVMR and MR BMA analyses that we used aim to pinpoint traits that have independent causal relationships with AMD, and the primary purpose of our sensitivity analysis was to deal with confounding. Given the hypothesis-free nature of our study, our analytical pipeline is intentionally not set up to interrogate biological pathways and to study interactions between exposures. Nonetheless it is clear that one area of significant interest for further study is the interaction between lipids and complement. Such an interaction is supported by the existing literature, which suggests that lipoproteins are related to complement activation. This is the focus of ongoing work by our group. Following the above comment, we have expanded the Discussion section and have included a mention in the conclusion to highlight that interaction between causal factors is an important area for future research (page 22, lines 528-532).

3. Consider expanding the discussion pertaining to traits found to be impactful in early but not advanced AMD. Might this reflect distinct causative factors within a subset of AMD prone to progression? Overall, the authors' don't seem to fully leverage the two datasets reflecting different stages of the same disease to make meaningful inferences about disease progression.

We agree with the reviewer that the early and advanced AMD datasets can be used to make inferences about disease progression. Our study design however involved conducting a primary analysis in an early AMD dataset, and then using an advanced AMD dataset as a form of validation. We opted for this analytical approach as it is understood that MR in elderly populations can be distorted by survival bias (*i.e.,* in an older population, the risk of bias increases because the analysis focusses on a non-random subset of the whole population who have lived sufficiently long to develop disease [PMID: 31373921]). Notably, the prevalence of advanced AMD increases with each decade of life and is highest after 75 years of age [PMID: 33569380]. Whilst all MR analyses of AMD risk are vulnerable to some extent of survival bias, the advanced AMD dataset is most vulnerable. Our motivation behind focussing on early AMD was therefore to produce the most statistically robust results; advanced AMD was then used exclusively as a form of validation (and to identify factors which are most likely relevant across the disease spectrum).

If one wanted to focus on exploring which factors are likely to cause individuals to progress from early to advanced AMD, it would be preferable to reverse the analysis with respect to the primary and secondary analysis. Specifically, the optimum approach would be to perform a phenome-wide causal inference study in advanced AMD and then observe which factors are significant in advanced AMD but not in early AMD. This analysis would have the potential to identify factors which account for the most severe cases of AMD. With that said, this form of analysis would be significantly more vulnerable to the biases which occur in an increasingly elderly population and would be weaker compared to our study with regards to identifying causal factors for AMD development. In summary, we agree with the reviewer that it would be interesting to explore which factors account for advanced AMD, and how these differ from early AMD. This however would require a separate study that would carefully consider survival bias.